# MeshSDF: Differentiable Iso-Surface Extraction

**Edoardo Remelli** [*1]    **Artem Lukoianov** [*1,2]    **Stephan R. Richter** [3]

**Benoît Guillard** [1]    **Timur Bagautdinov** [2]    **Pierre Baque** [2]    **Pascal Fua** [1]

[1]CVLab, EPFL, {name.surname}@epfl.ch
[2]Neural Concept SA, {name.surname}@neuralconcept.com
[3]Intel Labs, {name.surname}@intel.com

## Abstract

Geometric Deep Learning has recently made striking progress with the advent of *continuous* Deep Implicit Fields. They allow for detailed modeling of watertight surfaces of arbitrary topology while not relying on a 3D Euclidean grid, resulting in a learnable parameterization that is not limited in resolution.

Unfortunately, these methods are often not suitable for applications that require an *explicit* mesh-based surface representation because converting an implicit field to such a representation relies on the Marching Cubes algorithm, which cannot be differentiated with respect to the underlying implicit field.

In this work, we remove this limitation and introduce a differentiable way to produce explicit surface mesh representations from Deep Signed Distance Functions. Our key insight is that by reasoning on how implicit field perturbations impact local surface geometry, one can ultimately differentiate the 3D location of surface samples with respect to the underlying deep implicit field. We exploit this to define *MeshSDF*, an end-to-end differentiable mesh representation which can vary its topology.

We use two different applications to validate our theoretical insight: Single-View Reconstruction via Differentiable Rendering and Physically-Driven Shape Optimization. In both cases our differentiable parameterization gives us an edge over state-of-the-art algorithms.

## 1   Introduction

Geometric Deep Learning has recently witnessed a breakthrough with the advent of *continuous* Deep Implicit Fields [35, 29, 8]. These enable detailed modeling of watertight surfaces, while not relying on a 3D Euclidean grid or meshes with fixed topology, resulting in a learnable surface parameterization that is *not* limited in resolution.

However, a number of important applications require *explicit* surface representations, such as triangulated meshes or 3D point clouds. Computational Fluid Dynamics (CFD) simulations and the associated learning-based surrogate methods used for shape design in many engineering fields [3, 49] are a good example of this where 3D meshes serve as boundary conditions for the Navier-Stokes Equations. Similarly, many advanced physically-based rendering engines require surface meshes to model the complex interactions of light and physical surfaces efficiently [33, 36].

Combining explicit representations with the benefits of deep implicit fields requires converting the implicit surface parameterization to an explicit representation, which typically relies on one of

---

[*]Equal contribution

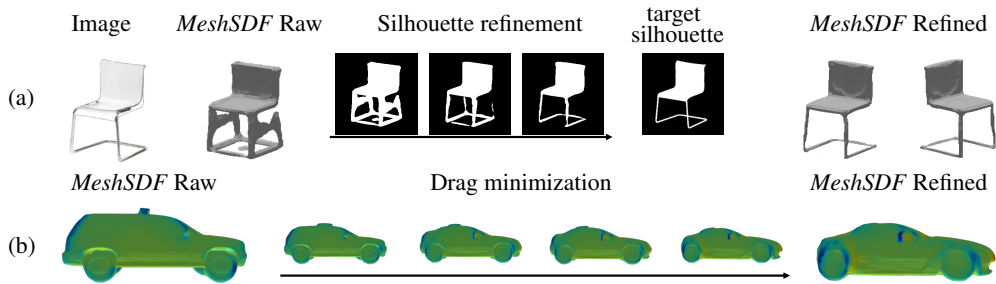

Figure 1: **MeshSDF**. (a) We condition our representation on an input image and output an initial 3D mesh, which we refine via differentiable rasterization [22], thereby exploiting MeshSDF's end-to-end differentiability. (b) We use our parameterization as a powerful regularizer for aerodynamic optimization tasks. Here, we start from an initial car shape and refine it to minimize pressure drag.

the many variants of the Marching Cubes algorithm [28, 32]. However, these approaches are not fully differentiable [24]. This effectively prevents the use of continuous Deep Implicit Fields as parameterizations when operating on explicit surface meshes.

The non-differentiability of Marching Cubes has been addressed by learning differentiable approximations of it [24, 51]. These techniques, however, remain limited to low-resolution meshes [24] or fixed topologies [51]. An alternative approach has been to reformulate downstream tasks, such as differentiable rendering [19, 26] or surface reconstruction [30], directly in terms of implicit functions, so that explicit surface representations are no longer needed. However, doing so is not easy and may even not be possible for more complex tasks, such as solving CFD optimization problems.

By contrast, we show that it is possible to use *continuous* signed distance functions to produce explicit surface representations while preserving differentiability. Our key insight is that 3D surface samples *can* be differentiated with respect to the underlying deep implicit field. We prove this formally by reasoning about how implicit field perturbations impact 3D surface geometry *locally*. Specifically, we derive a closed-form expression for the derivative of a surface sample with respect to the underlying implicit field, which is independent of the method used to extract the iso-surface. This enables us to extract the explicit surface using a non-differentiable algorithm, such as Marching Cubes, and then perform our custom backward pass through the extracted surface samples, resulting in an end-to-end differentiable surface parameterization that can describe arbitrary topology and is not limited in resolution. We will refer to our approach as *MeshSDF*.

We showcase the power and versatility of *MeshSDF* in the two different applications depicted by Fig. 1. First, we exploit end-to-end differentiability to refine Single-View Reconstructions through differentiable surface rasterization [22]. Second, we use our parameterization as powerful regularizer in physically-driven shape optimization for CFD purposes [3]. We will demonstrate that in both cases our end-to-end differentiable parameterization gives us an edge over state-of-the art algorithms.

In short, our core contribution is a theoretically well-grounded technique for differentiating through iso-surface extraction. This enables us to harness the full power of deep implicit surface representation to define an end-to-end differentiable surface mesh parameterization that allows topology changes.

## 2 Related Work

**From Discrete to Continuous Implicit Surface Models.** Level sets of a 3D function effectively represent watertight surfaces with varying topology [43, 34]. As they can be represented on 3D grids and thus easily be processed by standard deep learning architectures, they have been an inspiration for many approaches [5, 10, 13, 40, 42, 46, 52, 53]. However, methods operating on dense grids have been limited to low resolution volumes due to excessive memory requirements. Methods operating on sparse representations of the grid tend to trade off the need for memory for a limited representation of fine details and lack of generalisation [41, 42, 46, 47].

This has changed recently with the introduction of continuous deep implicit fields, which represent 3D shapes as level sets of deep networks that map 3D coordinates to a signed distance function [35] or occupancy field [29, 8]. This yields a continuous shape representation wrt. 3D coordinates that is lightweight but not limited in resolution. This representation has been successfully used for single view reconstruction [29, 8, 55] and 3D shape completion [9].

However, for applications requiring explicit surface parameterizations, the non-differentiability of iso-surface extraction so far has largely prevented exploiting the advantages of implicit representations.

**Converting Implicit Functions to Surface Meshes.** The Marching Cube (MC) algorithm [28, 32] is a widely adopted way of converting implicit functions to surface meshes. The algorithm proceeds by sampling the field on a discrete 3D grid, detecting zero-crossing of the field along grid edges, and building a surface mesh using a lookup table. Unfortunately, the process of determining the position of vertices on grid edges involves linear interpolation, which does not allow for topology changes through backpropagation [24], as illustrated in Fig. 2(a). Because this is a central motivation to this work, we provide a more detailed analysis in the Supplementary Section.

In what follows, we discuss two classes of methods that tackle the non-differentiability issue. The first one emulates iso-surface extraction with deep neural networks, while the second one avoids the need for mesh representations by formulating objectives directly in the implicit domain.

**Emulating Iso-Surface Extraction.** Liao et al. [24] map voxelized point clouds to a probabilistic topology distribution and vertex locations defined over a discrete 3D Euclidean grid through a 3D CNN. While this allows changes to surface topology through backpropagation, the probabilistic modelling requires keeping track of all possible topologies at the same time, which in practice limits resulting surfaces to low resolutions. Voxel2mesh [51] deforms a mesh primitive and adaptively increases its resolution. While this enables high resolution surface meshes, it prevents changes of topology.

**Reformulating Objective Functions in terms of Implicit Fields.** In [31], variational analysis is used to re-formulate standard surface mesh priors, such as those that enforce smoothness, in terms of implicit fields. Although elegant, this technique requires carrying out complex derivations for each new loss function and can only operate on an Euclidean grid of fixed resolution. The differentiable renderers of [20, 27] rely on sphere tracing and operate directly in terms of implicit fields. Unfortunately, since it is computationally intractable to densely sample the underlying volume, these approaches either define implicit fields over a low-resolution Euclidean grid [20] or rely on heuristics to accelerate ray-tracing [27], trading off in accuracy. 3D volume sampling efficiency can be improved by introducing a sparse set of anchor points when performing ray-tracing [25]. However, this requires reformulating standard surface mesh regularizers in terms of implicit fields using computationally intensive finite differences. Furthermore, these approaches [20, 25, 27] are tailored to differentiable rendering, and are not directly applicable to different settings that require explicit surface modeling, such as computational fluid dynamics.

# 3 Method

Tasks such as Single-View Reconstruction (SVR) [21, 17] or shape design in the context of CFD [3] are commonly performed by deforming the shape of a 3D surface mesh $\mathcal{M} = (V, F)$, where $V = \{\mathbf{v}_1, \mathbf{v}_2, ...\}$ denotes vertex positions in $\mathbb{R}^3$ and $F$ facets, to minimize a task-specific loss function $\mathcal{L}_{\text{task}}(\mathcal{M})$. $\mathcal{L}_{\text{task}}$ can be, e.g., an image-based loss defined on the output of a differentiable renderer for SVR or a measure of aerodynamic performance for CFD.

To perform surface mesh optimization robustly, a common practice is to rely on low-dimensional parameterizations that are either learned [4, 35, 2] or hand-crafted [3, 49, 39]. In that setting, a differentiable function maps a low-dimensional set of parameters $\mathbf{z}$ to vertex coordinates $V$, implying a fixed topology. Allowing changes of topology, an implicit surface representation would pose a compelling alternative but conversely require a *differentiable* conversion to explicit representations in order to backpropagate gradients of $\mathcal{L}_{\text{task}}$.

In the remainder of this section, we first recapitulate deep Signed Distance Functions, which form the basis of our approach. We then introduce our main contribution, a differentiable approach to computing surface samples and updating their 3D coordinates to optimize $\mathcal{L}_{task}$. Finally, we present *MeshSDF*, a fully differentiable surface mesh parameterization that can represent arbitrary topologies.

## 3.1 Deep Implicit Surface Representation

We represent a generic watertight surface $S$ in terms of a *signed distance function* (SDF) $s : \mathbb{R}^3 \rightarrow \mathbb{R}$. Given the Euclidean distance $d(\mathbf{x}, S) = \min_{\mathbf{y} \in S} d(\mathbf{x}, \mathbf{y})$ of a 3d point $\mathbf{x}$, $s(\mathbf{x})$ is $d(\mathbf{x}, S)$ if $\mathbf{x}$ is

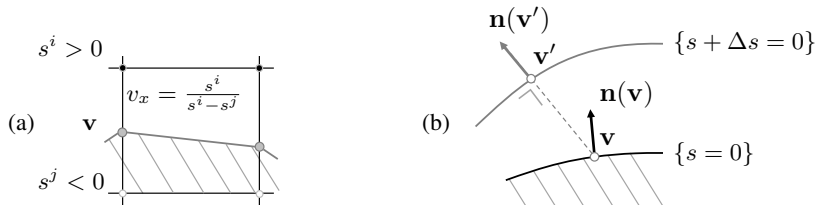

Figure 2: **Marching cubes differentiation vs Iso-surface differentiation.** (a) Marching Cubes determines the position $v_x$ of a vertex $\mathbf{v}$ along an edge via linear interpolation. This does not allow for effective back-propagation when topology changes because its behavior is degenerate when $s^i = s^j$ as shown in [24]. (b) Instead, we adopt a *continuous* model expressed in terms of how signed distance function perturbations locally impact surface geometry. Here, we depict the geometric relation between local surface change $\Delta \mathbf{v} = \mathbf{v}' - \mathbf{v}$ and a signed distance perturbation $\Delta s < 0$, which we exploit to compute $\frac{\partial \mathbf{v}}{\partial s}$ even when the topology changes.

outside $S$ and $-d(\mathbf{x}, S)$ if it is inside. Given a dataset of watertight surfaces $\mathcal{S}$, such as ShapeNet [6], we train a Multi-Layer Perceptron $f_\theta$ as in [35] to approximate $s$ over such set of surfaces $\mathcal{S}$ by minimizing

$$\mathcal{L}_{\text{sdf}}(\{\mathbf{z}_S\}_{S \in \mathcal{S}}, \theta) = \sum_{S \in \mathcal{S}} \frac{1}{|X_S|} \sum_{\mathbf{x} \in X_S} |f_\theta(\mathbf{x}, \mathbf{z}_S) - s(\mathbf{x})| + \lambda_{\text{reg}} \sum_{S \in \mathcal{S}} \|\mathbf{z}_S\|_2^2 , \tag{1}$$

where $\mathbf{z}_S \in \mathbb{R}^Z$ is the $Z$-dimensional encoding of surface $S$, $\theta$ denotes network parameters, $X_S$ represent 3D point samples we use to train our network and $\lambda_{\text{reg}}$ is a weight term balancing the contribution of reconstruction and regularization in the overall loss.

## 3.2 Differentiable Iso-Surface Extraction

Once the weights $\theta$ of Eq. 1 have been learned, $f_\theta$ maps a latent vector $\mathbf{z}$ to a signed distance field and the surface of interest is its zero level set. Recall that our goal is to minimize the objective function $\mathcal{L}_{\text{task}}$ introduced at the beginning of this section. As it takes as input a mesh defined in terms of its vertices and facets, evaluating it and its derivatives requires a *differentiable* conversion from an implicit field to a set of vertices and facets, something that marching cubes does not provide, as depicted by Fig. 2(a). More formally, we need to be able to evaluate

$$\frac{\partial \mathcal{L}_{\text{task}}}{\partial \mathbf{z}} = \sum_{\mathbf{v} \in V} \frac{\partial \mathcal{L}_{\text{task}}}{\partial \mathbf{v}} \frac{\partial \mathbf{v}}{\partial f_\theta} \frac{\partial f_\theta}{\partial \mathbf{z}} . \tag{2}$$

In this work, we take our inspiration from classical functional analysis [1] and reason about the *continuous* zero-crossing of the SDF $s$ rather than focusing on how vertex coordinates depend on the implicit field $f_\theta$ when sampled by the marching cubes algorithm. This results in a differentiable approach to compute surface samples $\mathbf{v} \in V$ from the underlying signed distance field $s$. We then simply exploit the fact that $f_\theta$ is trained to emulate a *true* SDF $s$ to backpropagate gradients from $\mathcal{L}_{\text{task}}$ to the underlying deep implicit field $f_\theta$.

To this end, let us consider a generic SDF $s$, a point $\mathbf{v}$ lying on its iso-surface $S = \{\mathbf{q} \in \mathbb{R}^3 | \ s(\mathbf{q}) = 0\}$, and see how the iso-surface moves when $s$ undergoes an infinitesimal perturbation $\Delta s$. Intuitively, $\Delta s < 0$ yields a local surface inflation and $\Delta s > 0$ a deflation, as shown in Fig. 2(b). In the Supplementary Section, we prove the following result, relating *local* surface change $\Delta \mathbf{v}$ to field perturbation $\Delta s$.

**Theorem 1.** *Let us consider a signed distance function $s$ and a perturbation function $\Delta s$ such that $s + \Delta s$ is still a signed distance function. Given such $\Delta s$, we define the associated local surface change $\Delta \mathbf{v} = \mathbf{v}' - \mathbf{v}$ as the displacement between $\mathbf{v}'$, the closest point to surface sample $\mathbf{v}$ on the perturbed surface $S' = \{\mathbf{q} \in \mathbb{R}^3 | \ s + \Delta s(\mathbf{q}) = 0\}$, and the original surface sample $\mathbf{v}$. It then holds that*

$$\frac{\partial \mathbf{v}}{\partial s}(\mathbf{v}) = -\mathbf{n}(\mathbf{v}) = -\nabla s(\mathbf{v}) , \tag{3}$$

*where $\mathbf{n}$ denotes the surface normals.*

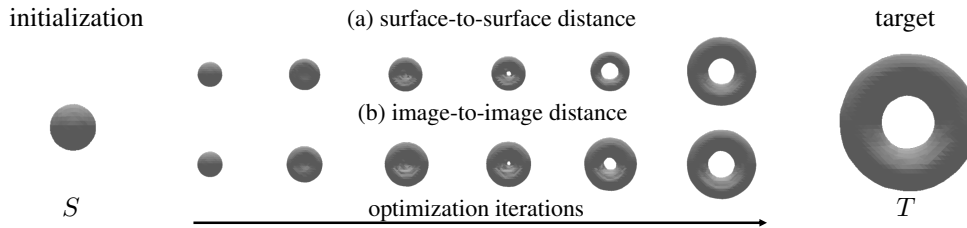

initialization  (a) surface-to-surface distance  target

(b) image-to-image distance

$S$  optimization iterations  $T$

Figure 3: **Topology-Variant Parameterization.** We minimize (a) a surface-to-surface or (b) an image-to-image distance with respect to the latent vector $\mathbf{z}$ to transform a sphere (genus-0) into a torus (genus-1). This demonstrates that we can backpropagate gradient information from mesh vertices to latent vector while modifying surface mesh topology.

Because $f_\theta$ is trained to closely approximate a signed distance function $s$, we can now replace $\frac{\partial \mathbf{v}}{\partial f_\theta}$ in Eq. 2 by $-\nabla f_\theta(\mathbf{v}, \mathbf{z})$, which yields

$$\frac{\partial \mathcal{L}_{\text{task}}}{\partial \mathbf{z}} = \sum_{\mathbf{v} \in V} -\frac{\partial \mathcal{L}_{\text{task}}}{\partial \mathbf{v}} \nabla f_\theta(\mathbf{v}, \mathbf{z}) \frac{\partial f_\theta}{\partial \mathbf{z}}(\mathbf{v}, \mathbf{z}) \, . \tag{4}$$

In short, given an objective function defined with respect to surface samples $\mathbf{v} \in V$, we can backpropagate gradients all the way back to the latent code $\mathbf{z}$, which means that we can define a mesh representation that is differentiable end-to-end while being able to capture changing topologies, as will be demonstrated in Section 4.

When performing a forward pass, we simply evaluate our deep signed distance field $f_\theta$ on an Euclidean grid, and use marching cubes (MC) to perform iso-surface extraction and obtain surface mesh $\mathcal{M} = (V, F)$. Conversely, we follow the chain rule of Eq. 4 to assemble our backward pass. This requires us to perform an additional forward pass of surface samples $\mathbf{v} \in V$ to compute surface normals $\nabla f_\theta(\mathbf{v})$ as well as $\frac{\partial f_\theta}{\partial \mathbf{z}}(\mathbf{v}, \mathbf{z})$. We implement *MeshSDF* following the steps detailed in Algorithms 1 and 2. Refer to the Supplementary Section for a detailed analysis of the computational burden of iso-surface extraction within our pipeline.

---

**Algorithm 1:** MeshSDF Forward

1: **input:** latent code $\mathbf{z}$
2: **output:** surface mesh $\mathcal{M} = (V, F)$
3: assemble grid $G_{3D}$
4: sample field on grid $S = f_\theta(\mathbf{z}, G_{3D})$
5: extract iso-surface $(V, F) = \text{MC}(S, G_{3D})$
6: **Return** $\mathcal{M} = (V, F)$

---

**Algorithm 2:** MeshSDF Backward

1: **input:** upstream gradient $\frac{\partial \mathcal{L}}{\partial \mathbf{v}}$ for $\mathbf{v} \in V$
2: **output:** downstream gradient $\frac{\partial \mathcal{L}}{\partial \mathbf{z}}$
3: forward pass $s_{\mathbf{v}} = f_\theta(\mathbf{z}, \mathbf{v})$ for $\mathbf{v} \in V$
4: $\mathbf{n}(\mathbf{v}) = \nabla f_\theta(\mathbf{z}, \mathbf{v})$ for $\mathbf{v} \in V$
5: $\frac{\partial \mathcal{L}}{\partial f_\theta}(\mathbf{v}) = -\frac{\partial \mathcal{L}}{\partial \mathbf{v}} \mathbf{n}$ for $\mathbf{v} \in V$
6: **Return** $\frac{\partial \mathcal{L}}{\partial \mathbf{z}} = \sum_{\mathbf{v} \in V} \frac{\partial \mathcal{L}}{\partial f_\theta}(\mathbf{v}) \frac{\partial f_\theta}{\partial \mathbf{z}}(\mathbf{v})$

---

## 4 Experiments

We first use a simple example to show that, unlike marching cubes, our approach allows for differentiable topology changes. We then demonstrate that we can exploit surface mesh differentiability to outperform state-of-the-art approaches on two very different tasks, Single View Reconstruction[1] and Aerodynamic Shape Optimization[2].

### 4.1 Differentiable Topology Changes

In the experiment depicted by Fig. 3, we used a database of spheres and tori of varying radii to train a network $f_\theta$ that implements the approximate signed function $s$ of Eq. 1. As a result, $f_\theta$ associates to a latent vector $\mathbf{z}$ an implicit field $f_\theta(\mathbf{z})$ that defines spheres, tori, or a mix of the two.

[1]main corresponding author: edoardo.remelli@epfl.ch
[2]main corresponding author: artem.lukoianov@epfl.ch

We now consider two loss functions that operate on explicit surfaces $S$ and $T$

$$\mathcal{L}_{\text{task1}} = \min_{s \in S} d(s, T) + \min_{t \in T} d(S, t) \ , \tag{5}$$

$$\mathcal{L}_{\text{task2}} = \|\text{DR}(S) - \text{DR}(T)\|_1 \ , \tag{6}$$

where $d$ is the point-to-surface distance in 3D [38] and DR is the output of an off-the-shelf differentiable rasterizer [22], that is $\mathcal{L}_{\text{task1}}$ is the surface-to-surface distance while $\mathcal{L}_{\text{task2}}$ is the image-to-image distance between the two rendered surfaces.

In the example shown in Fig. 3, $S$ is the sphere on the left and $T$ is the torus on right. We initialize the latent vector $\mathbf{z}$ so that it represents $S$. We then use the pipeline of Sec. 3.2 to minimize either $\mathcal{L}_{\text{task1}}$ or $\mathcal{L}_{\text{task2}}$, backpropagating surface gradients to the underlying implicit representation. In both cases, the sphere smoothly turns into a torus, thus changing its genus. Note that even though we rely on a deep signed distance function to represent our topology-changing surfaces, we did *not* have to reformulate the loss functions in terms of implicit surfaces, as done in [31, 20, 27, 25]. We now turn to demonstrating the benefits of having a topology-variant surface mesh representation through two concrete applications, Single-View Reconstruction and Aerodynamic Shape Optimization.

## 4.2 Single-View Reconstruction

Single-View Reconstruction (SVR) has emerged as a standardized benchmark to evaluate 3D shape representations [10, 11, 15, 50, 8, 29, 37, 14, 41, 56, 47]. We demonstrate that our method is straightforward to apply to this task and validate our approach on two standard datasets, namely ShapeNet [6] and Pix3D [45]. More results, as well as failure cases, can be found in the Supplementary material.

**Differentiable Meshes for SVR.** As in [29, 8], we condition our deep implicit field architecture on the input images via a residual image encoder [16], which maps input images to latent code vectors $\mathbf{z}$. These latent codes are then used to condition the architecture of Sec. 3.1 and compute the value of deep implicit function $f_\theta$. Finally, we minimize $\mathcal{L}_{\text{sdf}}$ (Eq. 1) wrt. $\theta$ on a training set of image-surface pairs. This setup forms our baseline approach, *MeshSDF* (raw).

To demonstrate the effectiveness of the surface representation proposed in Sec. 3.2, we exploit differentiability during inference via differentiable rasterization [22]. We refer to this variant as *MeshSDF*. Similarly to our baseline, during inference, the encoder predicts an initial latent code $\mathbf{z}$. Different to our baseline, our full version refines the predicted shape $\mathcal{M}$, as depicted by the top row of Fig. 1. That is, given the camera pose associated to the image and the current value of $\mathbf{z}$, we project vertices and facets into a binary silhouette in image space through a differentiable rasterization function $\text{DR}_{\text{silhouette}}$ [22]. Ideally, the projection matches the observed object silhouette $\mathcal{S}$ in the image, which is why we define our objective function as

$$\mathcal{L}_{\text{task}} = \|\text{DR}_{\text{silhouette}}(\mathcal{M}(\mathbf{z})) - \mathcal{S}\|_1 \ , \tag{7}$$

which we minimize with respect to $\mathbf{z}$. In practice, we run 400 gradient descent iterations using Adam [23] and keep the $\mathbf{z}$ with the smallest $\mathcal{L}_{\text{task}}$ as our final code vector.

**Comparative results on ShapeNet.** We report our results on ShapeNet [7] in Tab. 1. We compare our approach against state-of-the-art mesh reconstruction approaches: reconstructing surface patches [15], generating surface meshes with fixed topology [50], generating meshes from voxelized intermediate representations [14], and representing surface meshes using signed distance functions [56]. We used standard train/test splits along with the renderings provided in [56] for all the methods we tested. We evaluate on standard SVR metrics [47], which we define in the Supplementary Section. We report our results in Tab. 1. *MeshSDF* (raw) refers to reconstructions using our encoder-decoder architecture, which is similar to those of [29, 8], without any further refinement. Our full method, *MeshSDF*, exploits end-to-end differentiability to minimize $\mathcal{L}_{\text{task}}$ with respect to $\mathbf{z}$. This improves performance by at least $12\%$ over *MeshSDF* (raw) on all metrics. As a result, our full approach also outperforms all other state-of-the-art approaches.

**Comparative results on Pix3D.** Whereas ShapeNet contains only rendered images, Pix3D [45] is a test dataset that comprises real images paired to 3D models. We follow the evaluation protocol and metrics proposed in [45], which we detail in the supplementary material.

Table 1: **Single view reconstruction results on ShapeNet Core.** Exploiting end-to-end differentiability to perform image-based refinement allows us to outperform all prior methods.

| Metric | Method | plane | bench | cabinet | car | chair | display | lamp | speaker | rifle | sofa | table | phone | boat | mean |
|---|---|---|---|---|---|---|---|---|---|---|---|---|---|---|---|
| IoU ↑ | AtlasNet [15] | 20 | 13 | 7 | 16 | 13 | 12 | 14 | 8 | 28 | 11 | 15 | 14 | 17 | 15 |
| | Mesh R-CNN [14] | 24 | 25 | 17 | 21 | 21 | 21 | 20 | 15 | 32 | 19 | 26 | 26 | 26 | 23 |
| | Pixel2Mesh [50] | 29 | 32 | **22** | 25 | 27 | 27 | **28** | 19 | 40 | 23 | **31** | 36 | 32 | 29 |
| | DISN [56] | **40** | 33 | 20 | 31 | 25 | 33 | 21 | 19 | **60** | **29** | 25 | 44 | **34** | 30 |
| | MeshSDF (raw) | 32 | 32 | 19 | 30 | 24 | 28 | 20 | 18 | 45 | 26 | 24 | 48 | 28 | 28 |
| | MeshSDF | 36 | **38** | **22** | 32 | 28 | **34** | 25 | **22** | 52 | **29** | **31** | **54** | 30 | **32** |
| EMD ·$10^2$ ↓ | AtlasNett [15] | 6.3 | 7.9 | 9.5 | 8.3 | 7.8 | 8.8 | 9.8 | 10.2 | 6.6 | 8.2 | 7.8 | 9.9 | 7.1 | 8.0 |
| | Mesh R-CNN [14] | 4.5 | 3.7 | 4.3 | 3.8 | 4.0 | 4.6 | 5.7 | 5.1 | 3.8 | 4.0 | 3.9 | 4.7 | 4.1 | 4.2 |
| | Pixel2Mesh [50] | 3.8 | 2.9 | 3.6 | 3.1 | 3.4 | 3.3 | 4.8 | 3.8 | 3.2 | 3.1 | 3.3 | 2.8 | 3.2 | 3.4 |
| | DISN [56] | **2.2** | 2.3 | 3.2 | 2.4 | 2.8 | 2.5 | 3.9 | 3.1 | **1.9** | **2.3** | 2.9 | 1.9 | **2.3** | 2.6 |
| | MeshSDF (raw) | 3.3 | 2.5 | 3.2 | 2.2 | 2.8 | 3.0 | 4.2 | 3.5 | 2.6 | 2.7 | 3.1 | 1.9 | 2.9 | 3.0 |
| | MeshSDF | 2.5 | **2.1** | 3.0 | **2.0** | **2.4** | **2.4** | **3.2** | **2.9** | **1.9** | 2.4 | **2.7** | **1.7** | **2.3** | **2.5** |
| CD-$l_2$ · $10^3$ ↓ | AtlasNett [15] | 10.6 | 15.0 | 30.7 | 10.0 | 11.6 | 17.3 | 17.0 | 22.0 | 6.4 | 11.9 | 12.3 | 12.2 | 10.7 | 13.0 |
| | Mesh R-CNN [14] | 13.3 | 8.3 | 10.5 | 7.2 | 9.8 | 10.9 | 16.4 | 14.8 | 6.9 | 8.7 | 10.0 | 6.9 | 10.4 | 10.3 |
| | Pixel2Mesh [50] | 12.4 | 5.5 | 8.2 | 5.6 | 6.9 | 8.2 | **12.3** | **11.2** | 6.0 | 6.8 | **7.9** | 4.7 | 7.9 | 8.0 |
| | DISN [56] | **6.3** | 6.6 | 11.3 | 5.3 | 9.6 | 8.6 | 23.6 | 14.5 | 4.4 | **6.0** | 12.5 | 5.2 | **7.8** | 9.7 |
| | MeshSDF (raw) | 10.6 | 9.5 | 8.8 | 4.2 | 8.2 | 12.4 | 25.9 | 20.4 | 8.9 | 11.5 | 14.6 | 6.2 | 17.1 | 12.0 |
| | MeshSDF | **6.3** | **5.4** | **7.8** | **3.5** | **5.9** | **7.3** | 14.9 | 12.1 | **3.4** | 7.8 | 10.7 | **3.9** | 10.0 | **7.8** |

| Input | Pixel2Mesh [50] | DISN [56] | *MeshSDF* (Ours) |
|---|---|---|---|

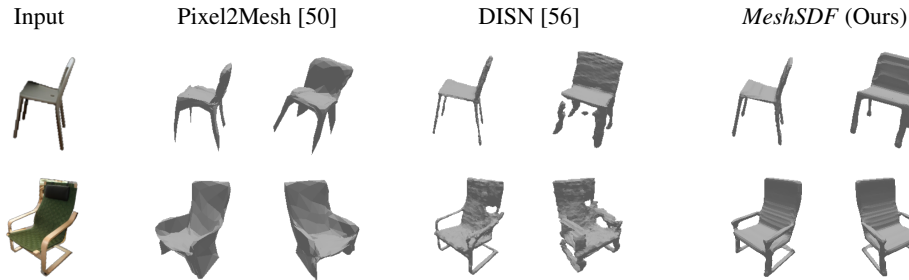

Figure 4: **Pix3D Reconstructions.** We compare our refined predictions to the runner-up approaches for the experiment in Tab. 2. *MeshSDF* can represent arbitrary topology as well as learn strong shape priors, resulting in reconstructions that are consistent even when observed from view-points different from the input one.

For this experiment we use the same function $f_\theta$ as for ShapeNet, that is, we do not fine-tune our model on Pix3D images, but train it on synthetic chair renders only so that to encourage the learning of stronger shape priors. We report our results in Tab. 2 and in Fig. 4. Interestingly, in this more challenging setting using real-world images, our simple baseline *MeshSDF* (raw) already performs on par with more sophisticated methods using camera information [56]. As for ShapeNet, our full model outperforms all other approaches.

Table 2: **Single view reconstruction results on Pix3D Chairs.** Our full approach outperforms all prior methods in all metrics.

| Metric | Pix3D [45] | AtlasNet [15] | Mesh R-CNN [14] | Pixel2Mesh [50] | DISN [56] | MeshSDF (raw) | MeshSDF |
|---|---|---|---|---|---|---|---|
| IoU ↑ | 0.282 | - | 0.240 | 0.254 | 0.333 | 0.337 | **0.407** |
| EMD ↓ | 0.118 | 0.128 | 0.125 | 0.115 | 0.117 | 0.119 | **0.098** |
| CD-$\sqrt{l_2}$ ↓ | 0.119 | 0.125 | 0.110 | 0.104 | 0.104 | 0.102 | **0.089** |

## 4.3 Shape Optimization

Computational Fluid Dynamics (CFD) plays a central role in designing cars, airplanes and many other machines. It typically involves approximating the solution of the Navier-Stokes equations using numerical methods. Because this is computationally demanding, *surrogate* methods [48, 54, 3, 49] have been developed to infer physically relevant quantities, such as pressure field, drag or lift, directly from 3D surface meshes without performing actual physical simulations. This makes it possible to optimize these quantities with respect to the 3D shape using gradient-based methods and at a much lower computational cost.

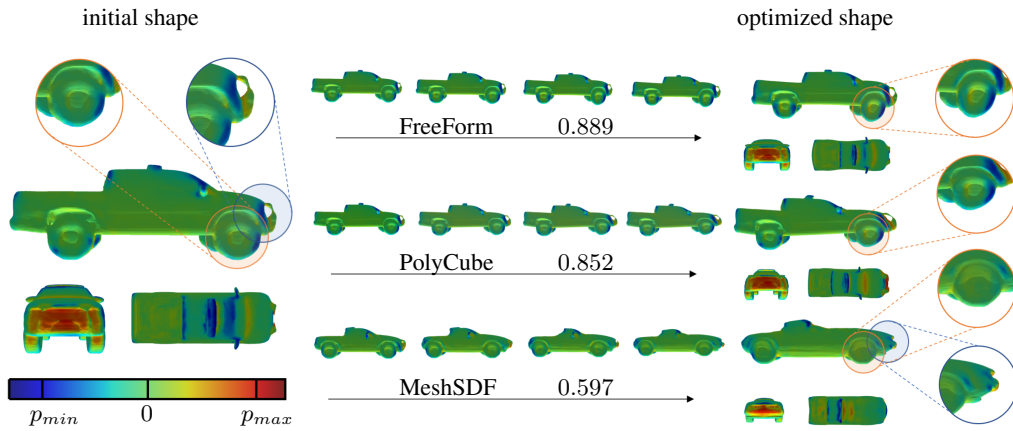

initial shape    optimized shape

FreeForm    0.889

PolyCube    0.852

MeshSDF    0.597

$p_{min}$    0    $p_{max}$

Figure 5: **Drag minimization.** Starting from an initial shape (left column), $\mathcal{L}_{task}$ is minimized using three different parameterizations: FreeForm (top), PolyCube (middle), and our *MeshSDF* (bottom). The middle column depicts the optimization process and the relative improvements in terms of $\mathcal{L}_{task}$. The final result is shown in the right column. FreeForm and PolyCube lack a semantic prior, resulting in implausible details such as sheared wheels (orange inset). By contrast, *MeshSDF* not only enforces such priors but can also effect topology changes (blue inset).

In practice, the space of all possible shapes is immense. Therefore, for the optimization to work well, one has to parameterize the space of possible shape deformations, which acts as a strong regularizer. In [3, 49] hand-crafted surface parameterizations were introduced. It was effective but not generic and had the potential to significantly restrict the space of possible designs. We show here that we can use *MeshSDF* to improve upon hand-crafted parameterizations.

**Experimental Setup.** We started with the ShapeNet car split by automatic deletion of all the internal car parts [44] and then manually selected $N = 1400$ shapes suitable for CFD simulation. For each surface $\mathcal{M}_i$ we ran OpenFoam [18] to predict a pressure field $p_i$ exerted by air travelling at 15 meters per second towards the car. The resulting training set $\{\mathcal{M}_i, p_i\}_{i=1}^N$ was then used to train a Mesh Convolutional Neural Network [12] $g_\beta$ to predict the pressure field $p = g_\beta(\mathcal{M})$, as in [3]. We use $\{\mathcal{M}_i\}_{i=1}^N$ to also learn the representation of Sec. 3.2 and train the network that implements $f_\theta$ of Eq. 1.

Finally, we introduce the aerodynamic objective function

$$\mathcal{L}_{task}(\mathcal{M}) = \iint_{\mathcal{M}} g_\beta \, \mathbf{n}_x \, d\mathcal{M} + \mathcal{L}_{constraint}(\mathcal{M}) \, , \qquad (8)$$

where the integral term approximates drag given the predicted pressure field, $\mathbf{n}_x$ denotes the projection of surface normals along airflow direction, and $\mathcal{L}_{constraint}$ is designed to preserve the required amount of space for the engine and the passenger compartment. Minimizing the drag of the car can now be achieved by minimizing $\mathcal{L}_{task}$ with respect to $\mathcal{M}$. We provide further details about this process and the justification for our definition of $\mathcal{L}_{task}$ in the Supplementary Section.

**Comparative Results.** We compare our surface parameterization to several baselines: (1) vertex-wise optimization, that is, optimizing the objective with respect to each vertex; (2) scaling the surface along its 3 principal axis; (3) using the *FreeForm* parameterization of [3], which extends scaling to higher order terms as well as periodical ones and (4) the *PolyCube* parameterization of [49] that deforms a 3D surface by moving a pre-defined set of control points.

We report quantitative results for the minimization of the objective function of Eq. 8 for a subset of 8 randomly chosen cars in Table 3, and show qualitative ones in Fig. 5. Not only does our method deliver lower drag values than the others but, unlike them, it allows for topology changes and produces semantically correct surfaces as shown in Fig. 5(c).

Table 3: **CFD-driven optimization**.We minimize drag on car shapes comparing different surface parameterizations. Numbers in the table (avg $\pm$ std) denote relative improvement of the objective function $\mathcal{L}_{\text{task}}^{\%} = \mathcal{L}_{\text{task}}/\mathcal{L}_{\text{task}}^{t=0}$ for the optimized shape, as obtained by CFD simulation in OpenFoam.

| Parameterization | None | Scaling | FreeForm [3] | PolyCube [49] | MeshSDF |
|---|---|---|---|---|---|
| Degrees of Freedom | $\sim 100k$ | 3 | 21 | $\sim 332$ | 256 |
| Simulated $\mathcal{L}_{\text{task}}^{\%} \downarrow$ | not converged | $0.931 \pm 0.014$ | $0.844 \pm 0.171$ | $0.841 \pm 0.203$ | $\mathbf{0.675 \pm 0.167}$ |

## 5 Conclusion

We introduce a new approach to extracting 3D surface meshes from Deep Signed Distance Functions while preserving end-to-end differentiability. This enables combining powerful implicit models with objective functions requiring explicit representations such as surface meshes. We believe that *MeshSDF* will become particularly useful for Computer Assisted Design, where having a topology-variant explicit surface parameterizations opens the door to new applications.

## 6 Acknowledgments

This project was supported in part by the Swiss National Science Foundation.

## 7 Broader Impact

Computational Fluid Dynamics is key to addressing the critical engineering problem of designing shapes that maximize aerodynamic, hydrodynamic, and heat transfer performance, and much else beside. The techniques we propose therefore have the potential to have a major impact in the field of Computer Assisted Design by unleashing the full power of deep learning in an area where it is not yet fully established.

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
