[Supplementary Material]

# MeshSDF: Differentiable Iso-Surface Extraction

**Edoardo Remelli** [*1]  **Artem Lukoianov** [*1,2]  **Stephan R. Richter** [3]

**Benoît Guillard** [1]  **Timur Bagautdinov** [2]  **Pierre Baque** [2]  **Pascal Fua** [1]

[1]CVLab, EPFL, {name.surname}@epfl.ch
[2]Neural Concept SA, {name.surname}@neuralconcept.com
[3]Intel Labs, {name.surname}@intel.com

## 1  Supplementary Material

In this supplementary material, we first remind the interested reader of why marching cubes are not differentiable and provide a formal proof of our main differentiability theorem. We then discuss our approach to speeding-up iso-surface extraction and performing end-to-end training. Finally, we give additional details about our experiments on single-view reconstruction and drag minimization.

### 1.1  Non-differentiability of Marching Cubes

Figure 1: **Marching cubes differentiation.** (a) Marching Cubes determines the relative position $x$ of a vertex **v** along an edge via linear interpolation. This does not allow for effective back-propagation when topology changes because of a singularity when $s_i = s_j$. (b) We plot $x$, relative vertex position along an edge. Note the infinite discontinuity for $s_i = s_j$.

The Marching Cubes (MC) algorithm [14] extracts the zero level set of an implicit field and represents it *explicitly* as a set of triangles. As discussed in the related work section, it comprises the following steps: (1) sampling the implicit field on a discrete 3D grid, (2) detecting zero-crossing of the field along grid edges, (3) assembling surface topology (i.e. the number of triangles within each cell and how they are connected) using a lookup table and (4) estimating the vertex location of each triangle by performing linear interpolation on the sampled implicit field. These steps can be understood as topology estimation followed by the determination of surface geometry.

More formally, let $S = \{s_i\} \in \mathbb{R}^{N \times N \times N}$ denote an implicit field sampled over a discrete Euclidean grid $G_{3D} \in \mathbb{R}^{N \times N \times N \times 3}$, where $N$ denotes the resolution along each dimension. Within each voxel, surface topology is determined based on the sign of $s_i$ at its 8 corners. This results in $2^8 = 256$ possible surface topologies within each voxel. Once topology has been assembled, vertices are created in case the implicit field changes sign along one of the edges of the voxel.

---

[*]Equal contribution

Figure 2: **Iso-surface differentiation.** We adopt a *continuous* model in terms of how small perturbations of a signed distance function locally impact surface geometry. Here, we depict the geometric relation between local surface change $\Delta \mathbf{v} = \mathbf{v}' - \mathbf{v}$ and a signed distance perturbation $\Delta s < 0$, which we exploit to compute $\frac{\partial \mathbf{v}}{\partial s}$ in the formal derivation below.

Specifically, the vertex location $\mathbf{v}$ is determined using linear interpolation. Let $x \in [0, 1]$ denote the vertex relative location along an edge $(\mathbf{G}_i, \mathbf{G}_j)$, where $\mathbf{G}_i$ and $\mathbf{G}_j$ are grid corners such that $s_j < 0$ and $s_i \geq 0$. This implies that if $x = 0$ then $\mathbf{v} = \mathbf{G}_i$ and conversely if $x = 1$ then $\mathbf{v} = \mathbf{G}_j$. In the MC algorithm, $x$ is is determined as the zero crossing of the interpolant of $s_i$, $s_j$, that is,

$$x = \frac{s_i}{s_i - s_j} \ . \tag{1}$$

Fig. 1(a) depicts this process. The vertex location is then taken to be

$$\mathbf{v} = \mathbf{G}_i + x(\mathbf{G}_j - \mathbf{G}_i). \tag{2}$$

Unfortunately, this function is discontinuous for $s_i = s_j$, as illustrated in Fig 1(b). Because of this, we cannot swap the signs of $s_i, s_j$ through backpropagation. This prevents topology changes while differentiating, as discussed in [12].

## 1.2 Proof of Differentiable Iso-Surface Result

Here we formally prove Theorem 1 from the main manuscript.

**Theorem 1.** *Let us consider a signed distance function $s$ and a perturbation function $\Delta s$ such that $s + \Delta s$ is still a signed distance function. Given such $\Delta s$, we define the associated local surface change $\Delta \mathbf{v} = \mathbf{v}' - \mathbf{v}$ as the displacement between $\mathbf{v}'$, the closest point to surface sample $\mathbf{v}$ on the perturbed surface $S' = \{\mathbf{q} \in \mathbb{R}^3 \mid s + \Delta s(\mathbf{q}) = 0\}$, and the original surface sample $\mathbf{v}$. It then holds that*

$$\frac{\partial \mathbf{v}}{\partial s}(\mathbf{v}) = -\mathbf{n}(\mathbf{v}) = -\nabla s(\mathbf{v}) \ , \tag{3}$$

*where $\mathbf{n}$ denotes the surface normals.*

*Proof.* Recalling the definition of signed distance field, from elementary geometry we have that

$$\Delta \mathbf{v} = \mathbf{v}' - \mathbf{v} = \mathbf{n}(\mathbf{v}')d(\mathbf{v}, S') = \mathbf{n}(\mathbf{v}')\left(s(\mathbf{v}) - \Delta s(\mathbf{v})\right) = -\mathbf{n}(\mathbf{v}')\Delta s(\mathbf{v}). \tag{4}$$

Now, observing that $\lim_{\Delta s \to 0} \mathbf{v}' = \mathbf{v}$, we have

$$\frac{\partial \mathbf{v}}{\partial s}(\mathbf{v}) = \lim_{\Delta s \to 0} \frac{\Delta \mathbf{v}}{\Delta s} = \lim_{\Delta s \to 0} -\mathbf{n}(\mathbf{v}') = -\mathbf{n}(\mathbf{v}). \tag{5}$$

Finally, recalling that for a signed distance field $\mathbf{n}(\mathbf{v}) = \nabla s(\mathbf{v})$, follows our claim. $\qquad \square$

Fig. 2 illustrates this proof.

## 1.3 Accelerating Iso-Surface Extraction

Recall that our approach to iso-surface differentiation method is independent from the technique used to extract surface samples, meaning that *any* non-differentiable iso-surface extraction method could be used to obtain an explicit surface from the underlying deep implicit field.

Figure 3: **Accellerated Iso-Surface extraction.** When working in an iterative optimization setting, we can exploit the fact that $f_\theta$, the implicit field underlying our surface mesh representation, will change only little between iterations to evaluate it *only* where we will expect it to change and consequently accelerate iso-surface extraction.

In practice, when operating in an iterative optimization settings such as those considered in the main manuscript, we exploit the fact that the deep implicit field $f_\theta$ is expected not to change drastically from one iteration to another, and re-evaluate it *only* where we can expect new zero-crossings to appear. In this setting, we evaluate $f_\theta$ only at grid corners where $|f_\theta|$ was smaller than a given threshold at the previous iteration. This reduces the computational complexity of field-sampling from $O(N^3)$ to $O(N^2)$ in terms of the grid size $N$, which brings noticeable speed ups, as illustrated in the benchmark of Fig. 3.

## 1.4 Comparison to Deep Marching Cubes

Deep Marching Cubes (DMC) [12] is designed to convert point clouds into a surface mesh probability distribution. It can handle topological changes but is limited to low resolution surfaces for the reasons discussed in related work. In the visualization below, we compare our approach to DMC. We fit both representations to a toy dataset consisting of two shapes: a genus-0 cow, and a genus-1 rubber duck. We use a latent space of size 2. Our metric is Chamfer $l_2$ distance evaluated on 5000 samples for unit sphere normalized shapes and shown at the bottom of the figure. As reported in the original paper, we found DMC to be unable to handle grids larger than $32^3$ because it has to keep track of all possible mesh topologies defined within the grid. By contrast, deep implicit fields are not limited in resolution and can better capture high frequency details.

| | DMC@$32^3$ | Ours@$32^3$ | Ours@$256^3$ | ground truth | DMC@$32^3$ | Ours@$32^3$ | Ours@$256^3$ | ground truth |
|---|---|---|---|---|---|---|---|---|
| CD-$l_2 \cdot 10^2$ ↓ | 1.87 | 1.84 | **1.80** | | 1.98 | 1.94 | **1.90** | |

## 1.5 End-to-End Training

Here, we demonstrate how our differentiable iso-surface extraction scheme can be used also to backpropagate gradient to the weights of MeshSDF, thus enabling end-to-end training. Specifically, let us consider a metric measuring the distance between two surfaces, such as the Chamfer $l_2$ distance

$$\mathcal{L}_{\text{chamfer}} = \sum_{\mathbf{p} \in P} \min_{\mathbf{q} \in Q} \|\mathbf{p} - \mathbf{q}\|_2^2 + \sum_{\mathbf{q} \in Q} \min_{\mathbf{p} \in P} \|\mathbf{p} - \mathbf{q}\|_2^2 \, , \tag{6}$$

where $P$ and $Q$ denote surface samples.

Figure 4: **Shilouette-driven refinment.** At inference time, given an input image, we exploit the differentiability of *MeshSDF* to refine the predicted surface so that to match input silhouette in image space through Differentiable Rasterization [10].

We exploit our differentiability result to compute

$$\frac{\partial \mathcal{L}_{\text{chamfer}}}{\partial \theta} = \sum_{\mathbf{v} \in V} \frac{\partial \mathcal{L}_{\text{chamfer}}}{\partial \mathbf{v}} \frac{\partial \mathbf{v}}{\partial f_\theta} \frac{\partial f_\theta}{\partial \theta} \tag{7}$$

$$= \sum_{\mathbf{v} \in V} -\frac{\partial \mathcal{L}_{\text{chamfer}}}{\partial \mathbf{v}} \nabla f_\theta \frac{\partial f_\theta}{\partial \theta} \ . \tag{8}$$

That is, we can train *MeshSDF* so that to minimize directly our metric of interest.

We evaluate the impact of doing so in Tab. 1, where we fine-tune *DeepSDF* models trained minimizing the loss function $\mathcal{L}_{\text{sdf}}$ of the main manuscript by further minimizing $\mathcal{L}_{\text{chamfer}}$. We refer to this variant as *MeshSDF*. Unsurprisingly, fine-tuning pre-trained models by minimizing the metric of interest allows us to obtain a boost in performance. In future work, we plan to pursue the following directions within end-to-end training: increasing the level of detail in the generated surfaces by exploiting Generative Adversarial Networks operating on surface mesh data [4], and train Single View Reconstruction architectures in a semi-supervised setting, that is by using *only* differentiable rasterization/rendering to supervise training.

Table 1: **End-to-end training.** We exploit end-to-end-differentiability to fine-tune pre-trained *DeepSDF* networks so that to that to minimize directly our metric of interest, Chamfer distance.

| Category | DeepSDF(train) | MeshSDF(train) | DeepSDF(test) | MeshSDF(test) |
|----------|----------------|----------------|---------------|---------------|
| Cars | 0.00071 | **0.00064** (↓ 9%) | 0.00084 | **0.00067** (↓ 20%) |
| Chairs | 0.00145 | **0.00133** (↓ 8%) | 0.00407 | **0.00259** (↓ 36%) |

## 1.6 Single View Reconstruction

We first provide additional details on the Single View Reconstruction pipeline presented in the main manuscript. Then, for each experimental evaluation of the main paper, we first introduce metrics in details, and then provide additional qualitative results. To foster reproducibility, we will make our entire code-base publicly available.

**Architecture.** Fig 4 depicts our full pipeline. As in earlier work [15, 3], we condition our deep implicit field architecture on the input images via a residual image encoder [8], which maps input images to latent code vectors **z**. Specifically, our encoder consists of a ResNet18 network, where we replace batch-normalization layers with instance normalization ones [22] so that to make harder for the network to use color cues to guide reconstruction. These latent codes are then used to condition the signed distance function Multi-Layer Perceptron (MLP) architecture of the main manuscript, consisting of 8 Perceptrons as well as residual connections, similarly to [18]. We train this architecture, which we dub *MeshSDF* (raw), by minimizing $\mathcal{L}_{\text{sdf}}$ (Eq.1 on the main manuscript) wrt. $\theta$ on a training set of image-surface pairs.

At inference time, we exploit end-to-end differentiability to refine predictions as depicted in Fig 4. That is, given the camera pose associated to the image and the current value of **z**, we project vertices and facets into a binary silhouette in image space through a differentiable rasterization function $DR_{\text{silhouette}}$ [10]. Ideally, the projection matches the observed object silhouette $\mathcal{S}$ in the image, which

is why we define our objective function as

$$\mathcal{L}_{\text{task}} = \|\text{DR}_{\text{silhouette}}(\mathcal{M}(\mathbf{z})) - \mathcal{S}\|_1 \,, \tag{9}$$

which we minimize with respect to $\mathbf{z}$. In practice, we run 400 gradient descent iterations using Adam [11] and keep the $\mathbf{z}$ with the smallest $\mathcal{L}_{\text{task}}$ as our final code vector.

**Evaluation on ShapeNet.** We used standard train/test splits along with the renderings provided in [25] for all the comparisons we report. We evaluate different approaches based on the following SVR metrics:

- **Chamfer $l_2$ pseudo-distance:** Common evaluation metric for measuring the distance between two uniformly sampled clouds of points $P, Q$, defined as

$$\text{CD-}l_2(P, Q) = \sum_{\mathbf{p} \in P} \min_{\mathbf{q} \in Q} \|\mathbf{p} - \mathbf{q}\|_2^2 + \sum_{\mathbf{q} \in Q} \min_{\mathbf{p} \in P} \|\mathbf{p} - \mathbf{q}\|_2^2. \tag{10}$$

  We evaluate this metric by sampling 2048 points from reconstructed and target shape, which are re-scaled to fit into a unit-radius sphere.

- **Earth Mover distance:** This metric measures the distance between two point clouds by solving an assignment problem

$$\text{EMD}(P, Q) = \min_{\Phi: P \to Q} \sum_{\mathbf{p} \in P} \|\mathbf{p} - \Phi(\mathbf{p})\|_2, \tag{11}$$

  where, for all but a zero-measure subset of point set pairs, the optimal bijection $\Phi$ is unique and invariant under infinitesimal movement of the points. In practice, the exact computation of EMD is too expensive and we implement the $(1 + \varepsilon)$ approximation scheme of [5]. We evaluate this metric by sampling 2048 points from reconstructed and target shape, which are re-scaled to fit into a unit-radius sphere.

- **Intersection over Union:** Since all information about an object's shape is situated on its surface, and to allow comparison to methods that do not produce watertight surfaces (such as [7]) we propose to evaluate object similarity by measuring surface-to-surface IoU. In practice, denoting as $\mathcal{V}$ the function mapping a cloud of points to a binary voxel grid, this metric reads

$$\text{IoU}(P, Q) = \frac{\text{intersection}(\mathcal{V}(P), \mathcal{V}(Q))}{\text{union}(\mathcal{V}(P), \mathcal{V}(Q))} \tag{12}$$

  We evaluate this metric by sampling 5000 points and setting up the voxel grid divide the object bounding box at resolution $50 \times 50 \times 50$.

- **F-score:** The F-Score has been recently proposed [21] for evaluating SVR algorithms. It explicitly evaluates the distance between object surfaces and is defined as the harmonic mean between precision and recall at a given distance threshold $d$. We refer the reader to [21] for more details about this metric. We evaluate this metric by sampling 10000 points from reconstructed and target shape and set 5% of the object bounding box length as distance threshold.

In Table 2, we further compare our method to state-of-the-art single view reconstruction algorithms in terms of F-score. Similarly to what reported in the main manuscript for CD, EMD and IoU, performing imaged-based refinement allows us to outperforms all other state-of-the-art approaches also in terms of this metric.

**Evaluation on Pix3D.** We followed closely the evaluation pipeline proposed together with this dataset [20]. That is, we focus on the chair category, and exclude from the evaluation all images where the object we want to reconstruct is truncated or occluded, resulting in 2894 test images. We then use ground truth bounding boxes to crop the image to a window centered around the object. To evaluate fairly reconstruction performance, we segment the background off for all methods presented in Table 2 of the main paper but for [20], that achieves state-of-the-art performance in joint segmentation and reconstruction on this benchmark. We do so to give a sense of the impact of assuming to have accurate segmentation information on reconstruction quality. Finally, following the evaluation

Table 2: **Single view reconstruction results on ShapeNet Core.** Exploiting end-to-end differentiability to perform image-based refinement allows us to outperform all prior methods also in terms of F-Score.

| Metric | Method | plane | bench | cabinet | car | chair | display | lamp | speaker | rifle | sofa | table | phone | boat | mean |
|---|---|---|---|---|---|---|---|---|---|---|---|---|---|---|---|
| F-Score% ↑ | AtlasNet | 91 | 86 | 74 | 94 | 91 | 84 | 81 | 80 | 96 | 91 | 91 | 90 | 90 | 89 |
| | Pixel2Mesh | 88 | 95 | **94** | 97 | 94 | 92 | 89 | 89 | 95 | **96** | 93 | 97 | 94 | 93 |
| | Mesh R-CNN | 87 | 91 | 90 | 95 | 90 | 89 | 83 | 85 | 93 | 92 | 90 | 95 | 91 | 90 |
| | DISN | 94 | 94 | 89 | 96 | 90 | 92 | 78 | 85 | 96 | **96** | 87 | 96 | 93 | 91 |
| | MeshSDF(raw) | 92 | 95 | 92 | 98 | 94 | 91 | 85 | 86 | 96 | 94 | 91 | 95 | 93 | 91 |
| | MeshSDF | **96** | **97** | **94** | **98** | **97** | **95** | **91** | **91** | **98** | **96** | **94** | **98** | **95** | **95** |

pipeline designed in [20], we only have access to ShapeNet synthetic data to train our models, that is we don't have access to any Pix3D image at training time. The main challenge of this benchmark is therefore to design an architecture that is robust to the change of domain. Finally, we use evaluation metrics as originally proposed in [20]:

- **Chamfer $\sqrt{l_2}$ pseudo-distance:**

$$\text{CD-}\sqrt{l_2}(P,Q) = \sum_{\mathbf{p}\in P} \min_{\mathbf{q}\in Q} \|\mathbf{p}-\mathbf{q}\|_2 + \sum_{\mathbf{q}\in Q} \min_{\mathbf{p}\in P} \|\mathbf{p}-\mathbf{q}\|_2, \tag{13}$$

  where $P$ and $Q$ are clouds of points. We evaluate this metric by sampling 1024 points from reconstructed and target shape, which are re-scaled to fit into a $[0.5, 0.5]^3$ bounding box.[2]

- **Earth Mover distance:** We use the same metric as above, but follow the approximation scheme of [20] in this case. We evaluate this metric by sampling 1024 points from reconstructed and target shape, which are re-scaled to fit into a $[0.5, 0.5]^3$ bounding box.

- **Intersection over Union:** We evaluate object similarity by measuring volume-to-volume IoU. In practice, denoting as $\mathcal{F}$ the function mapping a surface mesh $\mathcal{M}$ to a filled-in binary voxel grid, this metric reads

$$\text{IoU}_{\text{vol}}(\mathcal{P}, \mathcal{Q}) = \frac{\text{intersection}(\mathcal{F}(\mathcal{P}), \mathcal{F}(\mathcal{Q}))}{\text{union}(\mathcal{F}(\mathcal{P}), \mathcal{F}(\mathcal{Q}))}, \tag{14}$$

  where $\mathcal{P}, \mathcal{Q}$ denote surface meshes. We evaluate this metric by setting up the voxel grid divide the surface mesh bounding box at resolution $32 \times 32 \times 32$.

**Additional qualitative results.** We provide additional qualitative comparative results on both ShapeNet and Pix3D in Fig 8,9. Furthermore, in Fig 10 we show failure cases, which we obtain by selecting samples for which refinement does not bring any improvement. Furthermore, we refer the reader to the supplementary video for animations depicting the impact of iterative refinement on the reconstruction.

## 1.7 Aerodynamic Shape Optimization

Here we provide more details on how we performed the aerodynamic optimization experiments presented in the main manuscript. The overall pipeline for the optimisation process is depicted in Fig. 5, and additional optimization results are shown in Fig. 11.

### 1.7.1 Dataset

As described in the main manuscript, we consider the car split of the ShapeNet [2] dataset for this experiment. Since aerodynamic simulators typically require high quality surface triangulations to perform CFD simulations reliably, we (1) follow [19] and automatically remove internal part of each mesh as well as re-triangulate surfaces and (2) manually filter out corrupted surfaces. After that, we train a DeepSDF auto-decoder on the obtained data split and, using this model, we reconstruct the whole dataset from the learned parameterization. The last step is needed so that to provide fair initial conditions for each method of the comparison in Tab. 3 of the main manuscript, that is to allow all approaches to begin optimization from identical meshes.

Figure 5: **Aerodynamic optimization pipeline.** We encode a shape we want to optimize using *DeepSDF* (denoted as **SDF** block on the figure) and obtain latent code **z**. Then we start our iterative process. First, we assemble an Euclidean grid and predict SDF values for each node of the grid. On this grid we run the Marching Cubes algorithm (**MC**) to extract a surface mesh. We then run the obtained shape through a Mesh CNN (**CFD**) to predict pressure field from which we compute drag as our objective function. Using the proposed algorithm we obtain gradients of the objective w.r.t. latent code $z$ and do an optimization step. The loop is repeated until convergence.

Figure 6: **Simulated and predicted pressure fields.** Pressure fields for different shapes simulated with OpenFoam (top) and predicted by the Mesh Convolutional Neural Network (bottom).

We obtain ground truth pressure values for each car shape with OpenFoam [9], setting an *inflow velocity* of 15 meters per second and airflow *density* equal 1.18. Each simulation was run for at most 5000 time steps and took approximately 20 minutes to converge. Some result of the CFD simulations are depicted in the top row of Fig. 6.

We will make both the cleaned car split of ShapeNet and the simulated pressure values publicly available.

### 1.7.2 CFD prediction

We train a Mesh Convolutional Neural Network to regress pressure values given an input surface mesh, and then compute aerodynamic drag by integrating the regressed field. Specifically, we used the dense branch of the architecture proposed in [1] and replaced Geodesic Convolutions [16] by Spline ones [6] for efficiency.

A comparison for the predicted and simulated pressure values may be seen in Fig. 6.

### 1.7.3 Implementation Details

In this section we provide the details needed to implement the baselines parameterizations presented in the main manuscript.

Figure 7: **Soft constraints reserving space for driver and engine.** The figure illustrates the constraints we put on the surfaces during the optimization process. The constraints are shown for the initial shape, and then for all presented parameterizations. Note, that the constraints we put are soft, and thus may be violated.

- **Vertex-vise optimization** In this baseline, we optimize surface geometry by flowing gradients directly into surface mesh vertices, that is without using a low-dimensional parameterization. In our experiments, we have found this strategy to produce unrealistic designs akin to adversarial attacks that, although are minimizing the drag predicted by the network, result in CFD simulations that do not convergence. This confirms the need of using a low-dimensional parameterization to regularize optimization.

- **Scaling** We apply a function $f_{C_x,C_y,C_z}(V) = (C_x V_x, C_y V_y, C_z V_z)^T$ to each vertex of the initial shape. Here $C_i$ are 3 parameters describing how to scale vertex coordinates along the corresponding axis. As we may see from the Tab. 3 of the main manuscript, such a simple parameterization already allows to improve our metric of interest.

- **FreeForm** Freeform deformation is a very popular class of approaches in engineering optimization. A variant of this parameterization was introduced in [1], where it led to good design performances. In our experiments we are using the parameterization described in [1] with only a small modification: to enforce the car left and right sides to be symmetrical we square sinuses in the corresponding terms. We also add $l_2$-norm of the parameterization vector to the loss as a regularization.

- **PolyCube** Inspired by [23] we create a grid of control points to change the mesh. The grid size is $8 \times 8 \times 8$ and it is aligned to have $20\%$ width padding along each axis. The displacement of each control point is limited to the size of each grid cell, by applying $tanh$. During the optimization we shift each control point depending on the gradient it has and then tri-linearly interpolate the displacement to corresponding vertices. Finally, we enforce the displacement field to be regular by using Gaussian Smoothing ($\sigma = 1$, kernel size $= 3$). This results in a parameterization that allows for deformations that are very similar to the one of [23].

As we describe in the main paper, to prevent the surface from collapsing to a point, we put a set of soft-constraints to reserve space for driver and engine. The constraints are represented on the figure 7.

### 1.7.4 Additional Regularization for MeshSDF

In order to avoid generating unrealistic designs with MeshSDF, we introduce an additional regularization term $\mathcal{L}_{\text{constraint}}$ in the optimization, similarly to the regularizations introduced in the baseline parameterizations discussed above.

In our experiments, we began by using a standard penalty on $l_2$ norm of the latent code, $\mathcal{L}_{\text{constraint}} = \alpha||\mathbf{z}||_2^2$. However, even though it prevented most of the runs from converging to unrealistic shapes, we found converged shapes to still be coarse and noisy in some cases.

We therefore opted for a more conservative regularization strategy, reading

$$\mathcal{L}_{\text{constraint}} = \alpha \sum_{\mathbf{z}' \in \mathcal{Z}_k} \frac{||\mathbf{z} - \mathbf{z}'||_2^2}{|\mathcal{Z}_k|}, \tag{15}$$

where $\mathcal{Z}_k = \mathbf{z}_0, \mathbf{z}_1, \ldots, \mathbf{z}_k$ denote the $k$ closest latent vectors to $\mathbf{z}$ from the training set of DeepSDF. In our experiments we set $k = 10$, $\alpha = 0.2$. This regularization limits exploration of the latent space, but guarantees more robust and realistic optimisation outcomes.

In our aerodynamics optimization experiments, different initial shapes yield different final ones. We speculate that this behavior is due to the presence of local minima in the latent space of DeepSDF, even though we use the Adam optimizer [11] , which is known for its ability to escape some of them. We are planning to address the problem more thoroughly in future.

## 1.8 Comparison to implicit field differentiable rendering

Recent advances in differentiable rendering [13] have shown that is possible to render continuous SDFs differentiably by carefully designing a differentiable version of the sphere tracing algorithm. By contrast, we simply use MeshSDF end-to-end differentiability to exploit an *off-the-shelf* differentiable rasterizer to achieve the same result. To highlight the advantages of doing so, we take the generative model of Section 1.4, initialize latent code so that to generate the cow, and then minimize silhouette distance with respect to the duck. In the table below we compare our approach to [13]. Sphere tracing requires to query the network along each camera ray in a sequential fashion, resulting in longer computational time with respect to our approach, which projects surface triangles to image space and then rasterizes them in parallel. Furthermore, our approach requires less function evaluation, as we do not need to sample densely the volume around the field zero-crossing.

| Method | $l_2$ silhouette distance ↓ | # network queries ↓ | run time [s] ↓ |
|---|---|---|---|
| Liu20 [most efficient settings, $512^2$ renders] | 0.005973 | 898k | 1.24 |
| MeshSDF [isosurface at $256^3$, $512^2$ renders] | **0.004625** | **266k** | **0.29** |

## Footnotes

[2] In the main manuscript we have dubbed this metric as Chamfer $l_1$ by mistake. We will fix when we revise the paper.

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

Figure 10: **Failure cases for SVR on Pix3D.** Reconstruction refinement based on $L_1$ silhouette distance fails at capturing fine topological details for challenging samples. In the future, we plan to perform refinement using image-based loss functions that are more sensitive to topological mistakes [17].

Figure 11: **MeshSDF aerodynamic optimizations.**