[Reviews · NeurIPS 2020]

Review 1

Summary and Contributions: It is previously not differentiable to extract surface from deep sign distance field, e.g. by Marching Cube approaches. This paper presents a differentiable way, called MeshSDF. This open doors to the application of DeepSDF to various downstream tasks that require extracting surface.

Strengths: 1. First, the paper is well written and easy to follow. 2. The presented iso-surface differentiation is of high significance, both in terms of theory and applications. It is also well-grounded based on function analysis. 3. The differentiation is only applied in the backward pass and is independent of the way to sample the surface in the forward pass, therefore the method is generally applicable.

Weaknesses: I also have some confusions on current manuscript. 1. In Sec.3.2, after SDF changes a little bit, the new surface sample v' is defined as the sample on the new SDF that are closest to the original sample v. I wonder whether this is a principled definition or sort of heuristics? As far as my understanding, there is no constraint on the sample v that can be applied to sampling on both old and new SDF --- such constraint could establish the correspondence between v and v'. Thus, one could not find the v' corresponding to v unless some constraints are imposed, e.g. the closest point here. 2. Following the above definition of v' as the closet point, when we update z via backpropagation and run MC in the next iterations, would the new v' actually sampled by MC really the v' defined above? This is an important question to ask, because this is to make sure that the actually updating on v is guided by the computed differentiation. 3. Regarding minimizing equation (7), how to make the loss differentiable to the discrete values? Judging from line 201, DR_silhouette(M(z)) is a binary variable. I can imagine that when z is continuously changed, the DR_silhouette(M(z)) switches between 0 and 1 as a binary value and is thus a discrete function.

Correctness: The claims, method, and empirical methodology are correct.

Clarity: Yes

Relation to Prior Work: Yes

Reproducibility: Yes

Additional Feedback:


Review 2

Summary and Contributions: The authors address the task of deriving a surface mesh representation from a deep signed distance function with a differentiable iso-surface extraction approach. In contrast to applying Marching Cubes differentiation (where the position of a vertex along an edge is determined via linear interpolation and, hence, effective backpropagation is not possible for topology changes), iso-surface differentiation relies on a continuous model that describes how signed distance functions perturbations locally influence surface geometry. This allows a differentiable method for computing surface vertices from the signed distance field. In detail, the forward pass involves evaluating the signed distance field on a grid and using Marching Cubes to extract the iso-surface. The backward pass involves backpropagation of the gradients based on the chain rule (where a forward pass is used to get normals). The method’s potential is demonstrated in the scope of quantitative and qualitative results, however, some more results would strengthen the paper.

Strengths: Evaluation: - The authors evaluate that their approach can handle differentiable topology changes, which would not be possible with Marching Cubes. - The authors show the potential of their approach for single-view reconstruction based on quantitative and qualitative evaluations including a comparison to other methods. - In addition, they provide a practical application scenario for optimizing shapes in the scope of aerodynamic design. Exposition: The paper is well-written and easy to follow. Figures and captions are informative. In addition, the approach is reasonable. Reproducibility: The paper seems reproducible by the facts in the paper. In addition, the authors provide a detailed supplement with proofs regarding non-differentiability of marching cubes, the acceleration of iso-surface extraction as well as further details on the approach and its use for the different application scenarios.

Weaknesses: Evaluation: - A direct comparison to the results of Deep Marching Cubes, i.e. the differentiable Marching Cubes approach, would be interesting. As the authors also discuss in Section 2, we expect problems regarding the resolution, but the approach would also handle topological changes. - Figure 4 and Table 2 only demonstrate the performance for the category chairs. The visual evaluation would be strengthened by providing results for different types of objects. - A detailed discussion of failure cases and limitations is missing. - It’s not clear whether the change to a completely different car model/type is always the desirable solution (Figure 5). Would it be possible to constrain the underlying car type stronger? - A comparison of the computational burden and training times would be interesting for practical application. Exposition: - The value of the regularization strength lambda_reg in equation 1 seems not discussed. What value has been chosen and how susceptible is approach with respect to the respective parameter choice? Typos: line 70/194: wrt. -> w.r.t. line 127: represent -> represents in Table 1: AtlasNett -> AtlasNet line 221: so that to encourage line 260: Fig. 5c) -> there is no c) in Figure 5

Correctness: The paper seems correct and the method is reasonable as also verified in the results.

Clarity: The paper is well-written and easy to follow. The approach is well-motivated and figures and captions are informative. Furthermore, the information in the paper allow reproducing the technique.

Relation to Prior Work: The relation to the related Marching Cubes is sufficiently discussed, but the differentiable version Deep Marching Cubes seems to represent a relevant method to be considered for the comparison.

Reproducibility: Yes

Additional Feedback: see comments above Post-rebuttal comments: After reading the rebuttal and the other reviews, I am increasing my rating to accept. I would like to see the following revisions: - inclusion of the comparison to Deep Marching Cubes, as the increase in resolution is a clear benefit - inclusion of a comparison to differentiable rendering - discussion of limitations and failure cases in the main paper - in the main paper, a reference to the comparison of the computational burden and training times in the supplemental should at least be mentioned - an example better demonstrating the learning of an explicit mesh representation with variable topology would strengthen the evaluation (here, I agree with R3)


Review 3

Summary and Contributions: The paper proposes a way to differentiate through a (non-differentiable) marching cubes step when training deep implicit surfaces (e.g., DeepSDF, etc.). This makes it possible to optimize a loss function on the explicit mesh representation (as extracted using marching cubes) for a deep implicit surface, avoiding the usual topological constraints of mesh-based learning methods. The authors demonstrate their method experimentally on single view reconstruction using a differentiable mesh renderer and CFD-based optimization.

Strengths: State-of-the-art 3D deep learning methods typically must navigate the tradeoff between interpretable explicit parameterization (e.g., meshes) and variable topological structure (e.g., implicit surfaces). Typically, trying to combine the two induces a difficult combinatorial problem. The authors propose a novel way to circumvent this issue, taking advantage of spatial gradients of a learned implicit fields to allow the computation of "explicit" objective functions while learning an implicit representation. The experiments show convincing applications for why this is a useful capability, and, overall, the paper makes progress towards variable topology 3D learning.

Weaknesses: The theoretical derivative that is derived for a mesh vertex with respect to the implicit field is only valid under the assumption that the vertex is precisely on the zero level set of the implicit surface. Because of the discretization in marching cubes, this is not the case in the practice. It would be helpful to include some analysis, empirical and theoretical, about the stability of these gradients and the overall method with respect to the discretization. Relatedly, what grid resolution is used for marching cubes in the experiments, and how is this value determined? While this method enables the computation of mesh-based objectives without fixing surface topology, the representation that is ultimately learned is still an implicit function. For applications where mesh quality and resolution is important, there is no way to learn parameterizations that are better suited. Furthermore, it seems that in some cases, the use of loss functions on the explicit geometry obtained through marching cubes rather than on the implicit surface would actually negatively impact reconstruction due to the discretization. Is there a reasonable way to combine the explicit losses with some of the described implicit losses (used in DeepSDF, DISN, etc.)? It is also a bit surprising that without fine-tuning, ("MeshSDF Raw"), the proposed method consistently underperforms compared to the deep implicit function baseline (DISN). What is the reason for this? It would be interesting to see to what extent this is due to the inability to capture certain details of the predicted surfaces because of the marching cubes discretization.

Correctness: Besides some of the issues described under "Weaknesses" above, the theoretical and experimental components of the paper are well-designed.

Clarity: Overall the paper is clearly written and easy to follow. In equation 5, how are samples s \in S and t \in T for meshes S and T computed? Are they sampled randomly at each iteration? "AtlasNett" is misspelled in a few places.

Relation to Prior Work: Relevant methods are discussed and contextualized, and there is sufficient comparison to prior work.

Reproducibility: Yes

Additional Feedback: Post-rebuttal update: Thank you to the authors for the rebuttal. After reading the rebuttal and the other reviews, I'm keeping my original score. I agree that this is a nice a contribution and a step towards truly variable topology in 3D learning. I would love to see a bit more evidence that this method allows the use of explicit shape information during training, i.e., loss terms on the explicit geometry.


Review 4

Summary and Contributions: The paper introduces a new approach to extracting 3D surface meshes from Deep Signed Distance Functions while preserving end-to-end differentiability. To tackle the non-differentiability of traditional iso-surface extraction algorithm, e.g. Marching-Cubes, the paper derives a closed-form expression for the derivative of a surface sample with respect to the underlying implicit field. With the theoretically well-grounded technique for differentiating through iso-surface extraction, it's possible to train network end-to-end with loss over the mesh to generate arbitrary topology and unlimited resolution surface mesh.

Strengths: The technical contributions are sound. The evaluations and comparisons are thorough.

Weaknesses: The No.1 and No.3 experiments show that MeshSDF allows for differentiable topology changes and is applicable to shape optimization with explicit surface modeling. However, the No.2 experiment, single view reconstruction, is not sufficient. While all contrast works directly predict reconstructed mesh, the experiment only post-processes MeshSDF to refine reconstruction results. Another experiment is needed that based on a continuous implicit field-based single-view reconstruction method (e.g. the MeshSDF(raw) mentioned in the paper), various differential SDF surface extraction methods (such as differential rendering methods[1,2,3,4]) should be compared to refine reconstruction results with 2D silhouette loss. If MeshSDF cannot get much more impressive results in such an experiment, it's not convincing to prove that MeshSDF is better at harnessing the power of deep implicit surface representation. [1] DIST: Rendering Deep Implicit Signed Distance Function with Differentiable Sphere Tracing [2] Differentiable Volumetric Rendering: Learning Implicit 3D Representations without 3D Supervision [3] SDFDiff: Differentiable Rendering of Signed Distance Fields for 3D Shape Optimization [4] Learning to Infer Implicit Surfaces without 3D Supervision

Correctness: Correct.

Clarity: Very clear.

Relation to Prior Work: Yes, the DeepSDF and Deep Marching Cubes works are well discussed in the paper.

Reproducibility: Yes

Additional Feedback:

[Author Response · NeurIPS 2020]

We thank all reviewers for their thoughtful comments. Below, we address their concerns individually.

**[R1, closest point]** We don't impose any constraints on sampling point $\mathbf{v}$ and our theorem may be proven for any
mapping $\mathbf{v}'$ such that $\lim_{\Delta\mathbf{v}\to 0}(\mathbf{v}') = \mathbf{v}$. We chose the closest point one because it seemed the most natural to us.

**[R1, effective update]** The updated point indeed won't correspond to $\mathbf{v}'$ as described by the theorem because, when
back-propagating gradients to the latent code for refinement, we constrain surface change through a learned prior. This
is desirable, as we actually want surface deformations to be regularized by our learned shape-space.

**[R1, differentiable rasterization]** Rasterization is indeed not differentiable. We use the continuous relaxation of
[Kato18], approximating the discrete binary operation by a linear function to back-propagate gradients. We will clarify.

**[R2, comparison to DMC]** Deep Marching Cubes (DMC) is designed to convert point clouds into a surface mesh
probability distribution. It can handle topological changes but is limited to low resolution surfaces for the reasons
discussed in related work. In the figure below, we compare our approach to DMC. We fit both representations to a toy
dataset consisting of two shapes: a genus-0 cow, and a genus-1 rubber duck. We use a latent space of size 2. Our metric
is Chamfer $l_2$ distance evaluated on 5000 samples for unit sphere normalized shapes and shown at the bottom of the
figure. As reported in the original paper, we found DMC to be unable to handle grids larger than $32^3$ because it has to
keep track of all possible mesh topologies defined within the grid. By contrast, our approach is unlimited in resolution
   and can capture high frequency details, such as the ears of the cow.

| DMC@$32^3$ | Ours@$32^3$ | Ours@$256^3$ | ground truth | DMC@$32^3$ | Ours@$32^3$ | Ours@$256^3$ | ground truth |

CD-$l_2 \cdot 10^2 \downarrow$   1.87   1.84   **1.80**   1.98   1.94   **1.90**

**[R2, different categories]** In the main paper, we followed the Pix3D benchmark and reported qualitative results for
chairs only. However, we show results for several other ShapeNet categories in Fig. 8 of Supplementary.

**[R2, failure cases and limitations].** Failure cases for SVR are presented in Fig. 10 of the Supplementary material.
Furthermore, an important limitation is that the training loss of Equation 1 is insufficiently sensitive to topological
errors and this is something we are working on.

**[R2, car constraints].** The simplest way to restrict changes would be to either limit training data to a specific car type
or to increase the regularization weight discussed in section 1.6.4 of the Supplementary material. A more difficult
but more powerful approach would be to design constraints directly in terms of the mesh surface. We believe the
differentiability of MeshSDF makes this a practical proposition and this will be a topic for future research.

**[R2, performance]** In section 3 of the Supplementary material, we analyze the computational performance of our
differentiable iso-surface extraction pipeline. We did not report performance for other components of the full pipeline
(e.g. DeepSDF network, differentiable rasterization) because they are discussed in the original papers.

**[R2, exposition]** Following DeepSDF, we set $\lambda_{reg} = 10^{-3}$. We will add this to a revised version of the manuscript.

**[R3, Marching Cubes discretization].** This is indeed a valid concern, as our differentiation result only holds for
samples on the zero-crossing surface. In our experiments, we extract surface samples at $256^3$ resolution. This yields an
average SDF value of $10^{-5}$ for the samples. In practice, this is small enough to safely apply our differentiation result.

**[R3, combining explicit and implicit losses]** Our parameterization enables us to jointly exploit the advantages of
explicit and implicit representations: in experiment 4.2, we train our network by supervising for the implicit field while,
at inference time, we use explicit surface mesh losses to preform refinement.

**[R3, DISN]** Unlike DISN, which uses camera information to perform perceptual feature pooling, our baseline (MeshSDF
Raw) does not exploit camera information. We speculate that this is the reason for the performance gap.

**[R3, Equation 5]** We sample points uniformly with respect to surface area when computing point-to-surface distance.

**[R4, comparison to differentiable rendering]** Indeed, recent advances in differentiable rendering [Liu20] have
shown that is possible to render continuous SDFs differentiably by carefully designing a differentiable version of the
sphere tracing algorithm. By contrast, we simply use MeshSDF end-to-end differentiability to exploit an *off-the-shelf*
differentiable rasterizer and achieve the same result. To highlight the advantages of doing so, we take the generative
model in figure above, initialize latent code so that to generate the cow, and then minimize silhouette distance with
respect to the duck. In the table below we compare our approach to [Liu20]. Sphere tracing requires to query the network
along each camera ray in a sequential fashion, resulting in longer computational time with respect to our approach,
which projects surface triangles to image space and then rasterizes them in parallel. Furthermore, our approach requires
less function evaluation, as we do not need to sample densely the volume around the field zero-crossing. We refer the
reader to Section 3 of the Supplementary section for additional information on how we query our network.

| Method | $l_2$ silhouette distance ↓ | # network queries ↓ | run time [s] ↓ |
|---|---|---|---|
| Liu20 [most efficient settings, $512^2$ renders] | 0.005973 | 898k | 1.24 |
| MeshSDF [isosurface extraction at $256^3$, $512^2$ renders] | **0.004625** | **266k** | **0.29** |

[Meta-Review · NeurIPS 2020]

All reviewers were excited about this work. R2 provides in their revised review a fantastic list of requested changes for this paper in the camera ready, which we request for the final version of this paper. Please also address R3's concerns re: the precision with which your contributions are described in the paper.